Life-cycle traits in the demosponge Hymeniacidon perlevis in a land-based fish farm

Mercurio Maria 1
Longo Caterina caterina.longo@uniba.it 1
Pierri Cataldo 1
Cardone Frine 2
Corriero Giuseppe 1
Lazic Tamara 1
Zupa Walter 3
Carbonara Pierluigi 3
1 Department of Bioscience, Biotechnology and Environment, University of Bari , Bari , Italy
2 Department of Integrative Marine Ecology, Zoological Station “Anton Dohrn” , Naples , Italy
3 COISPA Tecnologia & Ricerca, Stazione Sperimentale per lo Studio delle Risorse del Mare , Torre a Mare (Bari) , Italy
Ereskovsky Alexander
Electronic publication date: 2023 Feb 23
Publication date: 2023
Volume: 11
Electronic Location ID: e14685
Received 2022 Jun 23; Accepted 2022 Dec 13
Copyright: ©2023 Mercurio et al.
Copyright year: 2023
Copyright holder: Mercurio et al.
License: This is an open access article distributed under the terms of the Creative Commons Attribution License, which permits unrestricted use, distribution, reproduction and adaptation in any medium and for any purpose provided that it is properly attributed. For attribution, the original author(s), title, publication source (PeerJ) and either DOI or URL of the article must be cited.
License URL: https://creativecommons.org/licenses/by/4.0/

Keywords: Hymeniacidon perlevis, Life-cycle, Growth performances, Lan-based fish farm

Funding: European Community, Life Environment funding program: Remedia-Life project LIFE16 ENV/IT/000343 Apulia Region PO FEAMP 2014-2020 Mis. 2.47: Approcci innovativi per una acquacoltura integrata e sostenibile 003/INA/20 This research was funded by the European Community, Life Environment funding program: Remedia-Life project (LIFE16 ENV/IT/000343) and by Apulia Region PO FEAMP 2014-2020 Mis. 2.47: Approcci innovativi per una acquacoltura integrata e sostenibile, project ID number 003/INA/20. There was no additional external funding received for this study The funders had no role in study design, data collection and analysis, decision to publish, or preparation of the manuscript.

==============================
Background

The demosponge Hymeniacidon perlevis is characterized by wide geographic distribution and great adaptability to numerous and highly variable climatic and hydrological conditions. Indeed, the species can colonize many different environments, including several unusual ones, such as concrete drainage conduits of a marine land-based fish farm plant. This research aimed to enhance existing knowledge on the reproductive cycle and growth performance of H. perlevis while also evaluating the impact of a controlled supply of trophic resources, wastewater flow and constant water temperature on these biological traits.

Methods

Specimens included in this one-year study inhabited drainage conduits of a land-based fish farm. The approach included measurements of sponge biomass and occurrence and abundance of reproductive elements across different seasons and environmental parameters, such as fish biomass, trophic resources, and wastewater flow. Sponge growth and reproductive elements, including oocytes, spermatic cysts, and embryos, were measured monthly in sponges positioned in the drainage conduit, thus with different trophic resources but with constant water temperature. Finally, we used generalized additive models to describe variables that contribute the most to the growth of sponges.

Results

Growth performance showed marked variations during the study period. The highest increase in sponge volume was observed from August/September to January/March. The volume of sponges was principally determined by the reduction of reared fish biomass and the increase of pellet amount and wastewater flow. Sponge specimens exhibited an active state during the entire study, as proven by the occurrence of recruits. However, sexual elements were only sporadically observed, thus not permitting the recognition of a true sexual cycle.

Discussion

The results of the present study confirmed that H. perlevis exhibits high flexibility and adaptability to the differential, and somewhat extreme, environmental conditions. Indeed, this species can live, grow and reproduce in the drainage conduits of the fish farm, where the species face constant darkness, water temperature and continuous nutritional supply. In such conditions, H. perlevis display an active state during the entire year, while avoiding stages of decline and long dormancy usually observed in wild populations. It seems plausible that stable environmental conditions induce an almost continuous sexual phase, probably under the control of endogenous factors. No asexual elements were detected, although it was impossible to exclude the contribution of asexual reproduction in the origin of the newly settled sponges, which were repeatedly detected throughout the study. The growth performance seemed linked to the fish farm conditions, thus providing useful indications on the best maintenance conditions for H. perlevis in land-based integrated multitrophic systems, where the species could be used for wastewater treatment.

Introduction

Porifera are pluricellular invertebrates of ancient origin characterized by a high plasticity. They are able to display special adaptive strategies at cellular, structural, and reproductive levels, thus adapting their life cycle to a wide range of environmental variations (Gaino, Manconi & Pronzato, 1995; Hill & Hill, 2002; Manconi & Pronzato, 2015).

The effects of environmental factors on sponges’ life cycles have been extensively studied (Witte, 1996; Whalan, Battershill & Nys, 2007; Gaino, Cardone & Corriero, 2010; Wahab et al., 2014); in particular, the water temperature is known to be one of the most important environmental drivers that can affect gametogenesis, sex ratios, and sexual reproductive output (Reiswig, 1973; Fell, 1976; Maldonado & Young, 1996; Usher et al., 2004; Ettinger-Epstein et al., 2007; Riesgo, Maldonado & Durfort, 2007; Whalan, Battershill & Nys, 2007; Maldonado & Riesgo, 2008; Ereskovsky et al., 2013; Wahab et al., 2014; Wahab et al., 2017; Lanna et al., 2015; Lanna et al., 2018; Shaffer et al., 2020).

The ability to adapt their life cycle to highly variable climatic and hydrological conditions seem particularly enhanced in sponges with wide geographic distribution, such as the case of demosponge Hymeniacidon perlevis (Montagu, 1814), one of the most common European sponge species (De Voogd et al., 2022).

Due to the wide systematic and molecular revisions, under the name Hymeniacidon perlevis are now accepted species that were differently named in the past, including H. perleve (Erpenbeck & Van Soest, 2002; Sun et al., 2007; Mahaut et al., 2013), H. sanguinea, H. caruncula (De Voogd et al., 2022), H. sinapium and H. heliophila (Turner, 2020). Genetic and morphological data, referred to Europe, Atlantic coasts of North and South America, Pacific coast of North America and Asia, support the occurrence of only one species. The literature data also reported some records from New Zealand, Southwest Africa, and the Pacific coast of South America, but these records are not confirmed by genetic analysis. Overall, H. perlevis can be effectively considered a cosmopolitan and invasive species (Juniper & Steele, 1969; Stone, 1970; Erpenbeck & Van Soest, 2002; Xue et al., 2004; Picton & Goodwin, 2007; Mahaut et al., 2013; Turner, 2020). In the Mediterranean Sea, it is distributed in eastern and western basins (Erpenbeck & Van Soest, 2002).

Hymeniacidon perlevis can colonise both hard and soft bottoms in shallow subtidal and intertidal zones (Cabioch, 1968; Stone, 1970; Erpenbeck & Van Soest, 2002; Corriero, 1990). It is also one of the most common benthic species inhabiting central Mediterranean lagoons (Longo et al., 2016; Longo et al., 2017), where it can survive prolonged periods of air exposure (Gaino, Cardone & Corriero, 2010) and high anthropogenic impacts (Gaino, Cardone & Corriero, 2010; Longo et al., 2016; Longo et al., 2017), showing peculiar adaptive strategies; indeed, this species exhibits a life cycle that lasts one year and consists of four stages, related to the seasonal variations: dormancy, resuscitation, bloom, and decline (Juniper & Steele, 1969; Stone, 1970; Gaino, Cardone & Corriero, 2010; Cao et al., 2007). During these stages, the sponge biomass increases or decreases according to the environmental conditions (Stone, 1970; Cao et al., 2007; Gaino, Cardone & Corriero, 2010; Zhang et al., 2010), with an optimal water temperature survival ranging between 10 and 20 °C, corresponding to spring and summer (Stone, 1970; Cao et al., 2007; Gaino, Cardone & Corriero, 2010; Zhang et al., 2010).

Hymeniacidon perlevis is a viviparous species that can shift from gonochorism to a successive hermaphroditism, in which oocytes precede spermatogenesis (Sarà, 1961; Stone, 1970; Diaz, 1973; Gaino, Cardone & Corriero, 2010). Oocytes are fertilised in the sponge body, where they give rise to ciliated parenchymella larvae (Stone, 1970; Diaz, 1973; Xue, Zhang & Zhang, 2009; Gaino, Cardone & Corriero, 2010). Although subjected to the influence of many hydrological parameters, sexual reproduction usually starts in late spring and ends in summer (Stone, 1970; Gaino, Cardone & Corriero, 2010). Asexual reproduction by fragmentation, supported by high regenerative properties, is also widespread and important for population survival and spreading (Gaino, Manconi & Pronzato, 1995), especially under adverse environmental conditions (Stone, 1970; Gaino, Cardone & Corriero, 2010).

The extreme adaptive capacity of H. perlevis permits the species to colonize many artificial environments, including marine land-based fish farming plants. At COISPA experimental fish farm in southern Italy (Torre a Mare, Apulia; https://www.coispa.it/), H. perlevis was detected in the drainage wastewater conduits of fish rearing tanks a year after the rearing experiment carried out at this farm (G. Corriero, 2014, unpublished data). This unusual discovery offered the opportunity to investigate the reproductive cycle and growth performance of H. perlevis over an entire year, correlate biological traits with the main physicochemical parameters, and evaluate the impacts of these drivers on H. perlevis life-cycle phases in a land-based fish farm. Moreover, the present study sought to improve existing knowledge about the effective environmental conditions for H. perlevis maintenance in land-based fish farms, which is especially important given the recent interest in the use of this sponge as a bioremediator of microbial load, including those potentially pathogenic (Fu et al., 2006; Longo et al., 2010; Longo et al., 2016; Longo et al., 2022; Zhang et al., 2010).

Materials & Methods

The Demospongiae Hymeniacidon perlevis, family Halichondriidae Gray, 1867 in the natural environment has a variable external morphology, ranging from small and thin crusts to large massive forms (>30 cm in maximum diameter). Usually, short papillae and small digitations emerge from the sponge surface. Colour varies from yellow-orange to red or pale green. Its skeletal architecture comprises spicules and spicule bundles made of slightly curved styles, often with a faint subterminal swelling tylote-like.

The population of H. perlevis here investigated inhabits the bottom of drainage wastewater conduit at COISPA marine experimental land-based fish farm (Torre a Mare, southern Italy; http://www.coispa.it). This conduit receives wastewater from fish farming tanks and conveys it into the sea.

Sponge specimens were detected for the first time in the drainage conduits of COISPA a year after the sponge rearing experiment (G. Corriero, 2014, unpublished data). In that instance, donor specimens collected from Mar Piccolo of Taranto (Italy) (Gaino, Cardone & Corriero, 2010) were cut into fragments and reared in dedicated tanks. Before the mentioned experiment, no specimens were ever observed, thus suggesting that their origin could be attributed to the reared sponges.

Environmental parameters

The considered drainage conduit consists of a cement canal 4.0 m long, 0.57 m wide, and 0.15 m deep. It is closed by walkable inspection protection, making this part of the conduit completely dark (Longo et al., 2022) (Fig. 1). As described by Longo et al. (2022), the water used in this land-based fish farm is sea-groundwater with the temperature, salinity and pH almost constant throughout the year (18 ± 0.5 °C, 34 ± 1‰, 7.09 ± 0.029) (Carbonara et al., 2020; Carbonara et al., 2021). Indeed, in coastal areas, there can be aquifers of marine origin; if sea water percolating into an aquifer passes through limestone, as in this case, the water circulating into the fish farm is a marine nutrient-free groundwater, without macroscopic organisms of the benthos (Awal, McDowall & Christie, 2016; Allan et al., 2009).

Figure 1 Plant Diagram of the drainage conduit and the sponge frames allocation.

Schematic drawing (not in scale) and real picture of the wastewater drainage conduit where the sponge Hymeniacidon perlevis lives (modified from Longo et al., 2022) with polyvinyl chloride (PVC) frames utilized for sponge volume measurement. In the red boxes picts of H. perlevis within a PVC frame and a specimen collected for analysis. The blue arrow indicates the wastewater flow direction.

The wastewater from fish tanks flows out through drainage conduits and passes through the area with sponges. Each tank has an overflow system and a tap on the bottom. Each tap tank is open daily to permit the release of accumulated food and faeces found at the tank bottom; approximately one-third of the tank volume is discharged daily. The water flow in the drainage conduit ranges between 15 and 28 L s−1 before and after the tank discharge, respectively.

Growth performance

Sponge growth was evaluated monthly from March 2018 to March 2019 using the digital photo-monitoring method and was expressed as volume (cm3). For this purpose, nine 20 × 20 cm polyvinyl chloride (PVC) frames were randomly positioned in the drainage conduit: three frames far from the wastewater inlet (named A, B and C; I group), three frames in the middle part (named D, E and F; II group), and three frames close to the wastewater inlet (named G, H and I; III group). Frame D was placed in an area without sponges to test colonization process and growth rate of recruits. For a detailed assessment of growth, each frame was divided into four 10 × 10 cm sub-squares (Fig. 1).

Photography of each PVC frame was performed monthly. The area occupied by sponges in each sub-square was detected using ImageJ software and was expressed in cm2.

Sponge volume (cm3) in each frame was calculated as the product of the total area occupied by sponges by the average thickness in the centre of each 10 × 10 cm sub-square, measured using a millimetre stick skewered in the sponge.

Reproductive cycle

From March 2018 to April 2019, ten randomly chosen sponge samples of approximately three cm3 were taken along the drainage conduit every month to estimate the relevance of sexual reproduction. Sponge samples were taken outside the frames used for growth assessment. Samples were fixed in 4% formaldehyde in filtered seawater for 24 h, repeatedly washed in the same buffer, demineralized with hydrofluoric acid, dehydrated in the crescent alcohol series, and paraffin-embedded. For histological observations, 7 µm sections were cut from paraffin blocks using Rotary One microtome and then processed following routinely used methodology for setting up stable preparations. The sections were stained with toluidine blue and observed under a light microscope (Olympus BH2) to determine the monthly percentage of specimens with oocytes, spermatic cysts, embryos and larvae, mean diameter (considering them spherical) and mean density (number mm−3) of reproductive elements. The quantitative evaluation of reproductive elements was carried out using the Abercrombie formula (1941) as suggested by Elvin (1976).

GAM analysis

To assess growth performances of H. perlevis, a generalized additive model (GAM) analysis was used to describe the most effective variables influencing the increase in sponge volume in specific environmental conditions of aquaculture facilities. The analysis was accomplished using sponge total volume at each sampling time as the response variable. Different trials were done using either volume data from all frames or volume data from each sampling position (frames furthest from the wastewater inlet: A, B and C; frames in the middle part: D, E and F; and frames closest to the wastewater inlet: G, H and I). The best modelling performances were obtained from merging data of G, H and I frames.

GAM analysis was performed using mixed gam computation vehicle (mgcv) library (Wood, 2017) in the R environment considering eight factors influencing sponge growth pattern. The first variable that was thought to be responsible for the sponge growth was the amount of food provided to reared fish. Due to the different feeding regimes adopted in the experimental fish farm, three food categories were considered: monthly amount of pellet, fresh food and total food. To include in the model useful variable in describing animals load of the fish farm, four other variables were initially considered: number of active fish tanks, total number of fishes hosted in the farm, corresponding fish biomass and monthly water volumes provided. Also, “month effect” was initially tested in the GAM model aiming to describe a temporal trend in the model.

The variance inflation factor analysis (VIF) was used to assess the presence of multicollinearity. Only covariates showing VIF values lower than 5 were retained in the analysis (Zuur, Ieno & Elphick, 2010), rejecting the others. Furthermore, a subsequent selection of the analysis was accomplished using a backwards stepwise approach. The covariates firstly included in the model were rejected following the following three criteria: (i) if the estimated degrees of freedom of the variable was close to 1; (ii) if the interval of confidence was zero; and (iii) if the generalized cross-validation (GCV) score (Gu & Wahba, 1991) decreased when the covariate was removed from the model. Among the different family distributions tested in the final analysis the Poisson family was chosen using the default logarithmic link function.

Results

Growth performance

Examined specimens showed a decline in their initial volume from March (overall mean ± S.E. 857.5 ± 129.03 cm3) to May (overall mean ± S.E. 583.86 ± 138.44 cm3), but it was particularly low in June (overall mean ± S.E. 541 ± 139.34 cm3). From June to August, sponges seemed in growth stasis (Fig. 2, Table 1). From August 2018 to March 2019, there was a general, although not constant, growth increase (Fig. 2). The highest growth performances were achieved in the area closest to the wastewater inlet (III group; overall mean volume ± S.E. 1,424.2 ± 64.08 cm3); indeed, sponges in this area, and particularly those in frames G (overall mean volume ± S.E. 1,720 ± 91.23 cm3) and I (overall mean volume ± S.E. 1,486 ± 88.78 cm3), had the greatest volume and grew the most (Table 1; Figs. 2, 3A–3B). The same trend was observed for these sponges’ thicknesses (frame G: overall mean ± S.E. 4.31 ± 0.22 cm; frame I: overall mean ± S.E. 3.72 ± 0.22 cm). On the contrary, in the intermediate (II group; overall mean ± S.E. 402.29 ± 56 cm3) and the area far from wastewater inlet (I group; overall mean ± S.E. 478.46 ± 43.4 cm3), examined individuals had smaller biomass and grew less over time. Sponges in frames B and C, for instance, showed no significative volume (overall mean ± S.E. 242.04 ± 42.34 cm3 and 623.42 ± 50.16 cm3, respectively) and thickness (overall mean ± S.E. 7.73 ± 1.19 cm3 and 17 ± 1.6 cm3, respectively) variations during the study. Frame D, positioned in the central part of the channel was colonized by H. perlevis after two months; after colonization, sponge grew relatively slowly (overall mean volume ± S.E. 91.48 ± 28.56 cm3). Several sponges disappeared during the study, and this particularly referred to the areas located in the intermediate part of the channel. Specimens in the frame F showed a slow biomass decrease between March (580 cm3) and September (311 cm3) and then totally disappeared in October, probably because of the accumulation of dry food that caused severe necrosis phenomenon (Fig. 3C). Sponges in the frame E showed some areas affected by necrosis and completely disappeared in March 2019. Sponges in the frame A (overall mean volume ± S.E. 91.48 ± 28.56 cm3) died in August but recolonized the frame in October and then quickly grew (Table 1).

Figure 2 Growth performance (volume in cm3± standard error) of H. perlevis considering sponge position in the drainage conduit.

Frames furthest from the wastewater inlet: I group; frames in the middle part: II group; frames closest to the wastewater inlet: III group.

Table 1 Variations of sponge biomass (cm3) within each sampling frame from March 2018 to March 2019.

Frame	Month	
	March ‘18	April ‘18	May ‘18	Jun ‘18	July ‘18	August ‘18	September ‘18	October ‘18	November ‘18	December ‘18	January ‘19	February ‘19	March ‘19	
A	450	450	310	290	290	0	0	482	1,000	920	770	890	600	
B	520	540	250	138	138	130	250	164	97	100	136	321.5	362	
C	950	830	500	530	530	410	500	497	538	524	577	819	899.5	
D	0	0	2	10	12.12	86	124	149	11.2	10	99	247	256	
E	820	910	550	500	427	527	628.5	681	736	920	806	986	0	
F	580	390	440	300	270	280	311	0	0	0	0	0	0	
G	1,550	1,450	1,300	1,250	1,270	1,810	1,890	1,760	2,040	1,920	1,780	2,110	2,230	
H	840	810	800	803.4	850	924	1,206	1346.5	1,443	1,386	1,155	1,297	1,033	
I	1,150	1,440	1,100	1,050	1,110	1,380	1,640	1,480	1,630	1,690	1,710	1,910	2,040	

Figure 3 Hymeniacidon perlevis in the drainage conduit of the land-based fish farm during the observation period.

(A) PVC frames G, H and I at the beginning of the observation positioned on the sponge showing a brownish color; (B) the same frames at the end of the observation period covered by sponge displaying an external cream color; (C) in the PVC frame F, H. perlevis shows a necrosis area (red arrow) next to the dry fish food residues (pellet, white arrow).

Sexual cycle

Reproductive elements were discontinuously detected over study period. In particular, oocytes (Fig. 4A) were found in July and January, while spermatic cysts (Fig. 4B) and embryos (Fig. 4C) were detected during four months (January, March, May, and September, and January, April, July, and October respectively). No larvae had been observed.

Figure 4 Hymeniacidon perlevis histological sections showing the reproductive elements.

Oocyte in the early phase of differentiation (A) (bar = 20 µm); Spermatic cyst in the choanosome (B) (bar = 25 µm); Embryo (C) (bar 50 = µm).

The frequency of specimens showing reproductive elements, their mean diameter and density are reported in Figs. 5A–5D. Small oocytes (27.5 ± 3.5 μm) were detected only in two samples but had low densities (3.2/mm3 ±0.2) (Fig. 5B); they show an irregular perimeter, a granular cytoplasm and a nucleolate nucleus, corresponding to the early phase of differentiation. The production of small spermatic cysts (58.2 ± 5.5 μm) involved 100% and 80% of the examined samples in March and September, respectively, but with low densities (6.3/mm3 ±1.5) (Fig. 5C); different cysts contained sperm in various phases of differentiation. Small embryos (162.5 ± 9.6 μm) were rarely found (four sponge samples, in total) and had low density values (2.3/mm3 ±0.4) (Fig. 5D). None of the samples showed the coexistence of oocytes and sperm cysts.

Figure 5 Monthly frequency, mean diameter and density of Hymeniacidon perlevis reproductive elements.

Frequency of fragments with oocytes, spermatic cysts and embryos (A); Mean diameter (±SD) and density (±SD) of oocytes (B), spermatic cysts (C) and embryos (D) per mm3 of sponge tissue.

Effect of the abiotic variables on the growth of the sponge

The VIF analysis showed the presence of collinearity (VIF > 5) among variables (see details in the Material and Method section) and some of them were hence rejected from the GAM analysis: month, total food, number of active tanks, and number of animals in tanks. Thus, the variables not correlated each other and retained in the tested models are: the fish biomass reared, the artificial food administered (pellet) and the water volume.

The best model describing increase in sponge volume used Poisson family distribution with the default logarithmic link function and fish biomass, amount of pellet and water volume as covariates, following here reported formula:

proximal.frame.vol ∼ α + fi (biomassi, pelleti) + water.voli + ɛi

where:

proximal.frame.vol is the response variable of the model and represents the volume of the proximal stations;

α is the intercept;

f is the bidimensional smoother combining the effect of the biomass and of the pellet provided to the fish tanks;

water.vol is the volume of the water filtered;

ɛ ∼ N(0, σ2) is the error term.

In Table 2 are reported models results (explained deviance, adjusted R2 and Akaike Information Criterion) using different response variables, accounting for the frames position. The model using intermediate.frames.vol as response variable was rejected due to its overfitting behaviours.

Table 2 Performance of the models (explained deviance, adjusted R2 and Akaike Information Criterion), describing the sponge growth pattern, using different response variables (volume of sponge), accounting for the frames position.

Response	Explained dev.	adj. R2	AIC		
All. frames. vol	77.70%	0.401	116,016		
Closest. frames. vol	87.90%	0.707	51,000	Best	
Intermediate. frames. vol	95.50%	0.881	5,760	Overfitting	
Furthest. frames. vol	64.60%	0.113	203,098		

The final model explains up to 87.9% of deviance and has an R2 of 0.70. The estimated degrees of freedom of estimated smoother and linear relationship were significantly different from zero (p-value < 0.05).

The residuals distribution was normal and around zero; it provided a good dispersion of fitted values against response in the plots, reported in Fig. 6, and the estimation of coefficients for each covariate was significant (p < 0.05).

Figure 6 Diagnostic plots of the model growth describing the relationship of variables (fish biomass reared; fish food–pellet; water volume) in the growth cycle of H. perlevis.

On the left is reported the dispersion of residuals around zero; on the right is reported the volume of sponge predicted (response) by model vs the observed (fitted) values (volume of sponge) plot.

Thanks to the partial effect of explanatory variables, reported in Fig. 7, the model showed that the sponge volume increase (considering merged data of the frames G, H and I) is mainly determined by the reduction of fish biomass and consequent increase of pellet amount provided to fish, together with the increase of the water flow in the drainage conduit.

Figure 7 Smoother and linear partial effect for the explanatory variables estimated from GAM model for sponge volume growth.

Left: surface described by the bidimensional spline of fish biomass reared and feed (pellet) (edf = 6.99); Right: partial linear effect of the water volume variable (water.vol).

Discussion

Hymeniacidon perlevis is one of the most common benthic demosponge species in the Mediterranean Sea that seems tolerant to a wide spectrum of environmental variations (Cabioch, 1968; Stone, 1970; Erpenbeck & Van Soest, 2002; Corriero, 1990). The present study investigated the impact of fish farm parameters, including fish biomass and feed, water temperature and wastewater flow, on the growth and reproduction of the sponge species. Such findings are important not just for understanding H. perlevis flexibility and adaptability but could also have implications when using this sponge in bioremediation processes.

One-year data indicated a slow biomass increase under-investigated conditions. The growth performances were mainly influenced by reared fish biomass, pellet amount, and wastewater flow. The highest increase in a sponge volume was observed in conditions of reduced fish biomass, increased pellet amount, and high wastewater flow; moreover, sponges occurring near wastewater inlet seemed to have higher growth rates. Under persistent environmental conditions, i.e., constant dark and water temperature (18 °C), H. perlevis exhibited an active state during the whole year of study. Occurrence of sexual elements, however, was only sporadic, thus not permitting the recognision of a true sexual cycle.

Referring to H. sanguinea (syn. of H. perlevis) in the Gulf of Napoli, Sarà (1961) indicated variable reproductive cycle with the highest concentration of oocytes and spermatic cysts in June, few sexual elements in July, August, and September, a certain degree of subsequent hermaphroditism, and unbalanced sex ratio with male overabundance (Table 3). By studying population living in Langstone Harbour (English Channel, southeast England), Stone (1970) reported no trace of larvae settlement despite high percentage of specimens (45%–100%) containing embryos in July and August. According to this study (Stone, 1970), population shows only two vital stages: bloom in five months (May–September) and decline in seven months (October–April) (Table 3). The sexual cycle of H. perlevis (under the name of H. caruncula) was also investigated by Diaz (1973) in Étang de Thau (France) where the species showed successive hermaphroditism in which oogenesis preceded spermatogenesis: the rise of water temperature in spring triggers onset of oogenesis, which lasts from March to May; spermatogenesis begins a month later and runs from April to June (Table 3). Cao et al. (2007) reported the presence of four vital stages in a population occurring in South China Sea (Dalian) (Table 3), with a seasonality similar to that observed in the Mediterranean population found at Mar Piccolo of Taranto (Northern Ionian Sea—South Italy) (Gaino, Cardone & Corriero, 2010). Reproductive timing of H. perlevis at Mar Piccolo of Taranto seems to last five months, from April to August (Gaino, Cardone & Corriero, 2010); according to the latter, sex ratio in this population is unbalanced, with a female overabundance; two hermaphroditic specimens were also observed. A group of specimens living close to the soft bottom and influenced by severe anoxic crises showed a significantly lower density of sexual elements than intertidal specimens subjected to moderate water movement. This population displayed four vital stages corresponding to the seasonal variations: dormancy in late autumn and winter, resuscitation in spring, bloom in summer, and decline in late summer and early autumn (Table 3).

Table 3 Metanalysis of the reproductive cycle and vital stage of H. perlevis in the Northern hemisphere. In table are reported monthly frequency of individuals with reproductive elements and information about the vital stage available in the literature.

	Reproductive elements	Jan	Feb	Mar	Apr	May	Jun	Jul	Aug	Sep	Oct	Nov	Dec	References	
Gulf of Napoli (Tyrrhenian Sea, South Italy)	SC					50%	85%		50%	25%				Sarà (1961)	
O						15%			50%				
E, L						7.5%							
Langstone Harbour (English Channel, south east England)	E							77%	82%	25%	3%			Stone (1970)	
Étang de Thau (Mediterranean French coast)	SC					x	x							Diaz (1973)	
O				x	x									
E, L						x	x							
Mar Piccolo of Taranto (Northern Ionian Sea—South Italy)	SC					33%	33%	11%						Gaino, Cardone & Corriero (2010)	
O				22%	55%	55%	44%	33%						
E, L					33%	55%	44%	33%						
Torre a Mare (South Italy) (COISPA experimental fish farm)	SC	10%		100%		10%				80%				Present paper	
O	10%						10%							
E	10%			10%			10%			10%				
Vital Stage	
Langstone Harbour		De	De	De	De	B	B	B	B	B	De	De	De	Stone (1970)	
Lingshui Bay (South China Sea, Dalian)		Do	Do	Do	R	R	B	B	B	De	De	De	Do	Cao et al. (2007)	
Mar Piccolo of Taranto		Do	Do	R	R	B	B	B	De	De	Do	Do	Do	Gaino, Cardone & Corriero (2010)	
Torre a Mare (COISPA experimental fish farm)		B	B	B	De	De	De	R	R	B	B	B	B	Present paper	
Notes.

SC Spermatic Cysts

O Oocytes

E Embryos

L Larvae

x data reported as qualitative observations

Vital stages Do Dormancy

R Resuscitation

B Bloom

De Decline

Although having monthly biomass variations, it seems that the sponges studied here do not have dormancy phase. Contrarily to the wild populations in which this phase typically occurs in winter, H. perlevis at the fish farm seemed to grow more in autumn and winter. The absence of typical four-stage life cycle is probably due to the constant water temperature (18 °C), following optimal temperature range (10–20 °C) for the vital stage maintenance (Stone, 1970; Cao et al., 2007; Gaino, Cardone & Corriero, 2010). Furthermore, constant water temperature appears responsible for the observed characteristics of sexual cycle. Riesgo et al. (2014) reported that reproduction and brooding modes are mainly determined by phylogeny but environmental factors can also influence reproductive traits (e.g., the timing of gametogenesis, frequency and quantity of reproductive propagules), and among these, water temperature is the most studied environmental driver (Reiswig, 1973; Fell, 1976; Maldonado & Young, 1996; Usher et al., 2004; Ettinger-Epstein et al., 2007; Riesgo, Maldonado & Durfort, 2007; Whalan, Battershill & Nys, 2007; Mercurio, Corriero & Gaino, 2007; Mercurio et al., 2013; Maldonado & Riesgo, 2008; Ereskovsky et al., 2013; Wahab et al., 2014; Wahab et al., 2017; Lanna et al., 2015; Lanna et al., 2018). In wild populations of H. perlevis, the onset of gametogenesis seems to be triggered by rapid spring changes of water temperature. At Mar Piccolo of Taranto, Gaino, Cardone & Corriero (2010) reported that the water temperature increase in April promotes female gamete differentiation, which precedes the presence of spermatic cysts by one month. A similar trend was also observed at the Étang de Thau, where gamete differentiation in spring was triggered by increased water temperature and the species showed successive hermaphroditism in which oogenesis preceded spermatogenesis (Diaz, 1973).

In the present study, the course of sexual cycle was different from that observed in wild populations, as it seemed that there was no seasonality in the specimens of H. perlevis. The occurrence of sexual elements was sporadic throughout almost entire year and, if compared to those of wild populations (Gaino, Cardone & Corriero, 2010) and the tissues had low gametes density. No asexual elements were observed, although the occurrence of asexual phenomenon cannot be excluded, as origin of the newly settled and repeatedly detected sponges was not ascertained. When subjected to constant values of several ecological parameters, including water temperature, salinity, pH, and dark conditions (Carbonara et al., 2019; Carbonara et al., 2020; Carbonara et al., 2021), it seems plausible that H. perlevis have constant but not very productive sexual phase controlled by endogenous factors. According to the literature, endogenous processes, using a biological clock, are involved in the control of an obligate set of life cycle phases, but the seasonality and length of phases are subjected to exogenous factors (Simpson & Fell, 1974; Hill & Hill, 2002; Manconi & Pronzato, 2015; Lanna et al., 2021).

Examined sponge specimens showed marked growth variations during the study period. The applied model showed that the increase in sponge volume was mainly determined by the reduction of fish biomass and increase of available pellet amount, together with the increase of water flow in the drainage conduit that would avoid pellet stagnation in its central part.

Zhang et al. (2010) reported that H. perlevis reared in an intensive mariculture water system displayed growth, stationary, and decline stages during the test period. At the end of the experiment, the sponge biomass was lower than that at the beginning. The same trend has also been observed in a previous laboratory study (Zhang et al., 2005). Similar results were obtained in applied research performed at Mar Piccolo of Taranto (Bioremediation in aquaculture activities through the use of benthic invertebrates—FEP Project 39/OPI/010), demonstrating that the biomass of H. perlevis reared in vertical structures along the water column increased in the first months of rearing, but underwent a notable decrease due to the substrate competition with other filter-feeder organisms (C. Longo, 2014, unpublished data). Data collected in the present research showed that the total sponge biomass has slightly increased compared to the beginning of the observations. Overall, sponges located in the area closest to the wastewater inlet had the highest growth performances throughout the survey year. The sponges positioned in the central area showed evident signs of necrosis until their total disappearance, probably because of large quantities of accumulated solid food and a decrease in the water flow rate, which caused food stagnation and suffocation. The sponges located farthest from the water inlet area showed a low growth rate, probably due to the lower food availability mostly captured by the sponges placed upstream.

Compared to other groups, specimens in the frames closest to the wastewater inlet (group III) were subjected to a higher microbial concentration (Longo et al., 2022), thus having a greater quantity of easily filterable food. It is reasonable to hypothesize that the furthest frames (group I) received wastewater partially filtered by sponges in the previous frames (group III and II) with a lower microbial concentration.

The results implemented existing knowledge on the plastic and adaptable life cycle of H. perlevis in controlled systems while also providing valuable indications regarding species rearing exploitation in land-based fish farms. These findings open new perspectives on sponge use in land-based integrated multitrophic systems for waste treatment, which is especially important when considering H. perlevis’s ability to remove microbial pollutants, including those potentially pathogenic (Fu et al., 2006; Longo et al., 2010; Longo et al., 2016; Longo et al., 2022; Zhang et al., 2010).

Conclusions

The present research, conducted in drainage conduit of a marine land-based fish farm and characterized by constant darkness, water temperature and wastewater nutritional supply, indicated that demosponge H. perlevis maintains vital physiological state during the entire year. Contrary to what was observed in wild populations, H. perlevis in this fish farm avoided stages of decline and long dormancy. Low density of gametes and occasional presence of sexual elements did not permit the recognition of seasonality in the sexual cycle. It seems reasonable, however, that constant values of basic ecological parameters led to the almost persistent sexual phase controlled by endogenous factors. Lack of naturally occurring changes in water temperature in spring seem not allowing the onset of real sexual cycle as normally occurring in wild populations.

The sponge showed marked growth variations during the study period, with the highest biomass increase linked to the variations of some fish farming conditions, and this particularly referred to the reduction of reared fish biomass, increase of available pellet amount and wastewater flow. Due to peculiar life cycle characteristics and ability to remove microbial pollutants, the present research enhanced the existing knowledge while highlighting the opportunities for the use of H. perlevis in future land-based integrated multitrophic aquaculture systems for wastewater treatment in which this species could have an essential role.

Supplemental Information

Supplemental Information 1 Hymeniacidon perlevis volume data

Click here for additional data file.

Supplemental Information 2 Hymeniacidon perlevis reproduction data

Click here for additional data file.

Supplemental Information 3 Fishes feeding data

Click here for additional data file.

The authors would like to acknowledge the time and effort devoted by reviewers to improving the quality of the manuscript. We sincerely appreciate all valuable comments and suggestions, which helped us to improve the quality of the published manuscript.

Additional Information and Declarations

Competing Interests

Author Contributions

Data Availability

The authors declare there are no competing interests.

Maria Mercurio conceived and designed the experiments, performed the experiments, analyzed the data, prepared figures and/or tables, authored or reviewed drafts of the article, and approved the final draft.

Caterina Longo conceived and designed the experiments, analyzed the data, authored or reviewed drafts of the article, and approved the final draft.

Cataldo Pierri conceived and designed the experiments, performed the experiments, authored or reviewed drafts of the article, and approved the final draft.

Frine Cardone performed the experiments, analyzed the data, authored or reviewed drafts of the article, and approved the final draft.

Giuseppe Corriero conceived and designed the experiments, prepared figures and/or tables, authored or reviewed drafts of the article, and approved the final draft.

Tamara Lazic analyzed the data, authored or reviewed drafts of the article, and approved the final draft.

Walter Zupa analyzed the data, prepared figures and/or tables, authored or reviewed drafts of the article, and approved the final draft.

Pierluigi Carbonara conceived and designed the experiments, analyzed the data, prepared figures and/or tables, authored or reviewed drafts of the article, and approved the final draft.

The following information was supplied regarding data availability:

The raw data are available in the Supplemental Files.

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
