# Peer review of "Life-cycle traits in the demosponge Hymeniacidon perlevis in a land-based fish farm"

_PeerJ, doi:10.7717/peerj.14685_

## Round 0.1 · original submission · Major Revisions

Dear Dr. Longo,

I am agreeing with the critical comments of your interesting paper. Please carefully study all their comments and make appropriate corrections in the new version of the manuscript.

Best regards,
Alexander

·

Basic reporting

Dear Dr. Ereskovsky,

The manuscript entitled “Life-cycle traits in the demosponge Hymeniacidon perlevis living in a land-based fish farm” by Mercurio et al. submitted to PeerJ aimed to describe the growth pattern and the reproductive strategy of the widespread demosponge H. perlevis in a tank where fishes are grown for aquaculture. The authors took the opportunity of a water body with controlled conditions to investigate the effect of different environmental variables on these two life-history traits. The study was carried out in a narrow and shallow drainage conduit through which the wastewater of the tanks is removed. The sponges were transplanted to this region in a previous study and are being kept alive for a while there. They investigated both life cycles for one year and associated only the growth dynamics with some environmental factors. They found that the sponges grow at different rates depending on their position along the drainage channel, but found a good model to explain the dynamics at the farthest position in the channel. Finally, they found that this small population seems to reproduce without seasonality and with a very mild reproductive effort. They, then, compare their findings with other previous published in the literature.
In general, the text is fluid and easy to read. However, I found some words that are likely bad translations from Italian to English (I am pointing them out in the annotated pdf). However, overall, the text is well written and understandable by an international audience.
Although I like the approach and can understand the importance of such a study, I am afraid that the current version of the manuscript should be thoroughly and deeply reviewed before being considered for publication. I think that the authors have a very good question in hand, but it is not clearly demonstrated in the manuscript. For example, there is no clear statement of the objective of the work anywhere in the text. The methodology is fine, but some issues are not clearly stated (see specific comments below). The results are also fine, but the figures and tables should be reviewed to reach a scientific publication standard, as some images are blurry, graphs are direct outputs from Excel with colors that can confuse the reader, and the Table is not clearly prepared. There is much emphasis on the statistical output (numbers, actually) but little relevance to the biological meaning of the test. Finally, the discussion could better fit the objectives of the work and discuss how the environment seems to regulate and promote the growth and reproduction of this species. The data obtained in the current work is not really linked to the ones previously published.
I also think that the authors should draw an important line when comparing their current results with the previous publications. Although the species is clearly invasive in other areas (after Turner, 2020) and has a large resilience to different environmental conditions, the previous studies were carried out in the wild with natural populations. After their introduction, it is known that the studied small population is artificial and lives in a very different condition when compared to the other populations. Therefore, it is important to present the pros of the study when dealing with aquaculture issues, but also to highlight the caveats when comparing to the natural populations previously investigated. I mean, your work can be useful to both areas of science (aquaculture AND sponge biology), but you have to emphasize the pros and cons of the methodology applied here to them.
The authors provided three Excel files with spreadsheets containing the raw data, which could allow other researchers to repeat their analyses.
I believe that the authors can improve their work with my (many) comments along the text. Please, refer to the attached PDF file with my comments.

Experimental design

In general, the methods are well written and – apparently – repeatable. However, I think that the authors do not describe how they analyzed the data about the reproduction of the sponge. In the methods, the reader has to assume that they used the histological slides only to check the presence/absence of the reproductive elements. Can’t you quantify the reproductive elements on the slides, taking into consideration the method suggested by Lanna et al. (2018 – Hydrobiologia)? If the authors have this information, they could carry out the same statistical analysis that they used for the growth dynamics of the reproductive data and have more robust results on this topic.

The statistical analysis used in the study is interesting and can give very good results, as indicated by the high percentage of variation explained by the chosen model. However, I am afraid that the use of this model is very limited, especially in the way it is described. The authors do not explain why they choose to use only the data about the sponges located in the farthest point of the inlet. Moreover, they describe that the sponges found in each section of the drainage conduit had different growth patterns during the investigation. Therefore, how can one agree that the abiotic variables used to describe the dynamics of the growth of the sponge is representative of the species? I think that they need to model the average growth performance of the population in all plots rather than each plot separately. Moreover, I think that they also have to show the competing models. I suggest that they read the book by Burnhan and Anderson (2011) about model selection. There is nothing wrong with the method used here, but I feel that other factors could also explain – at some degree – the fluctuations in the size of the sponges.

About the reproduction analyses: do you collect fragments of the same sponges along the study or were they randomly chosen? The “fragments” are labeled 1-10 in the Excel supplementary file.

Validity of the findings

The work has a potential of a good impact, as they are describing some life-history traits of a sponge growing in aquaculture tanks that could alleviate the impact of the high biomass of fishes onto the wild environments nearby. They are providing a model that indicates what are the resources necessary to make the sponges grow better in controlled conditions. However, as I described above, I have some criticism about the real use of this model, as it is modeling only a portion of the studied sponges in a very small area. Conclusions, therefore, are poorly supported by the current data presented in the manuscript.

Additional comments

As this species was investigated previously by several authors, I believe that the authors could take this as an opportunity to present the gaps about the reproductive and growth biology of H. perlevis that is still not known. They are not presenting this in the Introduction.

In the Results session, the variation of the abiotic factors from the fish farm throughout the study should be informed. I think that figures showing the dynamics of the factors during the study will be very important to better understand how these factors and the growth dynamics of the sponge are related to each other. The authors say that the abiotic factors (e.g. temperature, salinity, oxygen) are stable along the year, but do not show any data (or citation) to support this affirmation. I believe that the work would be benefited from such addition.

I think that pictures showing the spermatic cysts, oocytes, and embryos of the species would be interesting, to describe the reproduction of the species.

Finally, there are many comments and suggestions to change the text along the annotated PDF attached to my comments.

I hope that my review encourages the authors to improve their work.
All the best,
Emilio

Reviewer 2 ·

Basic reporting

The manuscript presenting the life cycle and growth patterns of Hymeniacidon perlevis by Mercurio et al. contains interesting information about the variations that the reproductive cycle suffers when the conditions are stable throughout the year. It also measures the growth patterns along one year taking into account different variables related to the use of the fish farm that could potentially be exploited for sponge farming. Although, as I said, it contains relevant information for the field, the manuscript seems to me a very preliminary draft that can be largely improved by taking more measures, performing more analyses, rewriting some sections and putting a lot more care in the imaging side of the article. In particular, the results should be completely rewritten to be more succinct, using the support of good tables and figures and not detailing the measurements for every sponge, but in turn, focusing on the trends. Also, there is a bit of confusion between sections and they sometimes appeared mixed.

Experimental design

Introduction
Perhaps the word primitive does not really depict the reality of the sponges. I think the authors referred here to their ancient origin, and perhaps rewording the sentence would be better.

The term “bloom” is perhaps misleading in this context. It would be better if the authors refer to this period as reproduction.

Methods
Ovocites is not the correct term. Please use oocyte.

The entire paragraph about the genetic data supporting a single species should be moved to the introduction.

It should be said what parameters of the reproductive elements were measured, only presence / absence? If so, I suggest the authors take advantage of the work done at the histological level and measure the size of the reproductive elements and identify the developmental stage as well to provide the manuscript with more appropriate information.

Line 162. Food and feces instead of feed and faces.

Results
Growth performance
It is perhaps too detailed to talk about the individual growth of the sponges. I would suggest to use average and standard deviation measures, and keep the raw data for a table.

Reproduction
What is the size of each examined section? Please include this information.
What is the size of the spermatic cysts? In this sense, 7-10 spermatic cysts does not necessarily mean low density appearance if they are big.

What do you mean by fragment? Sample, individual? Please use the word sample or individual instead of fragment, because it is misleading and the reader could imagine the authors looking at different fragments of the same individual, and this was not the case.

Where the oocytes observed in the same vitellogenic stage? What about the spermatic cysts? And the embryos?

It would be desirable that the authors present images of the reproductive elements and report on the sizes of each class.

Here the results are mixed with discussion about previous studies. Please move that to the Discussion section.

Conclusions
The first paragraph recapitulated the discussion again, and can be removed. In fact, the conclusions are just repeated sentences from the discussion, and the entire section is not necessary.

Figures
In general the figures are poor quality, most times unfocused, and also the graphs look very amateur done in excel. I would suggest the authors to keep only a couple of pictures of the sponges were the individuals are visible (it is almost impossible to see the sponges in most of them) and perhaps one showing the necrotic individuals, and then take some good quality images of the reproductive elements.
For the graphs, I would suggest a more elegant presentation using ggplots in R for all graphs, and the use of averages/standard deviation.
Figure 1 is worse than the explanation in the text and can be removed.

Validity of the findings

The findings are relevant, but I encourage the authors to perform a much more thorough analysis of their histological sections to better understand the reproductive cycle and its modifications. (see my comments above).

Additional comments

Please check the spelling of Hymeniacidon throughout the manuscript.

---

## Round 0.2 · Minor Revisions

Dear Dr Longo,

I am agreeing with the critical comments of Reviewer 1 on your second version of the manuscript. Please carefully study all his comments and make appropriate corrections in the new version of the manuscript.

Best regards,
Alexander

·

Basic reporting

The manuscript entitled "Life-cycle traits in the demosponge Hymeniacidon perlevis in a land-based fish farm" by Mercurio and collaborators has been much improved since the first submission. I think that the authors have achieved most of the suggestions that other reviewers and I presented and the work is way better now.
The English is fine, with only very minor changes suggested below. References are sufficient, up-to-date, and essential for the context of the work.
Figures, Tables, and legends can still be improved (see below), but the manuscript's structure is well defined and good.
The results and hypotheses are well depicted and also fine.
Therefore, the manuscript is nice and well-prepared.

Experimental design

No comments.

Validity of the findings

No comments.

Additional comments

Dear Dr. Ereskovsky,
Dear authors,

I find the new version of the manuscript much better than the first version reviewed some time ago. I think that the authors worked very hard to cope with the comments of the other reviewers and my own, and am happy to recommend its publication after considering some of my following comments:

Line 270 - I think this session is too 'statistic' rather than biology-focused. I mean, the authors describe in quite detail how they perform the tests, which variables were used or left aside, and so on. In my opinion, the focus should be more related in the biology than the statistics per se. I would rather see all the 'details' of the test in the methods, as the test is the way in which you figure out that lower fish biomass, higher pellet abundance and water flow impact the growth of the sponge. It is OK to show the equation, indicate how adjusted this model is, and compare it with the other models. However, please focus on the relationship rather than on the statistics. This is not an obligatory change. It is just a perspective. It is up to Dr. Ereskovsky and other reviewers to ask for a proper modification.

Line 322-323 - I believe you are referring to your study, right?
Maybe use "According to the present study"
or
"In the current study"
or
"Here, we found that the population..."
You wrote "According to this study (Stone, 1970)" lines above, and it can be confusing.

Line 356-357 - The information about sex distribution is repeated here. It was already presented in the last paragraph. Please, organize the text to make it more fluid.

Line 364 - Please, consider changing the text to: ", the" in "Gaino et al., 2010) and tissues" >> Gaino et al., 2010), the tissues"

Line 395-397 - Another issue that came to my mind is the volume of water passing above the sponges. If this volume is large, even if the speed of the water passing in the conduit is slow, I am afraid that the filtration dynamics of the sponges would not be sufficient to leave very few food for the next group...
Anyways, just think about these possibilities and consider than to change the text, if you think it is necessary.

Line 413-414 - This conclusion appears here, but was not discussed earlier. Either you add it to the Discussion above, or remove it from the conclusions. It makes no sense to conclude something that was not mentioned before. You mention during the text that the rerproduction is based on endogenous characteristics. The conclusion should be this information, only.

My main concerns are still related to the figures and their subtitles. I think that the quality of the figures should be improved with sharper lines, higher definition pictures and even better organization of the parts in the figures are desirable. Legends can also be improved.

In figure 1 legend, what does "picts" mean?
In figure 3, I think that part C could be put to the right of part B. I also suggest the authors to change the font of the characters to a sans serif font (Helvetica or Arial, for instance). Finally, the most important thing about this figure: I am troubled to recognize what is sponge, what is substrate, what is other organisms... Is the sponge the white blots on the ground? Or the dark? Please, identify them in the figure. It can be very straightforward for you who spent a whole year looking at them in situ and in the photographs, but even for me - someone working with sponges - it is hard to figure out what is the sponge, imagine people who are not familiarized with these organisms?

Figure 4 - Please, check my comment about the font of the characters (A, B, and C) above.
An arrow pointing to the oocytes and another to the spermatic cysts are also desirable. Remember that some readers might not be familiarized with the reproductive elements of the sponges...

Figure 5 - The label of the parts of the figures (A-D) is not presented in the figure. Either you delete the "call" from the text and the legend (instead of calling Fig 5A, for instance, only call Fig 5) or you add the label in the figure. I'd rather see the second option.

Figure 6 - This figure, and its legend, reinforce my concerns about the focus on the statistical test rather than on the biology investigated here. What is this GAM analysis intended for? What is this residual about? Again, the way in which this result is shown demonstrates a high weight for the 'statistical method' rather than the biological importance of the test. These residual plots indicate that the model is very nicely fitted. Great! However, with the legend presented here, no one can understand how this is related to the growth and reproductive cycle of the sponges in this fish farm. At least, inform what these plots are related to. Maybe: "Diagnostic plots of the model describing the relationship of variable A, B, and C in the growth cycle of H. perlevis ..."

Figure 7 - the legend of this figure also needs to be reviewed having in mind the focus on the biology investigated in the paper. Moreover, it should be more complete. For instance, 'biomass' of what?
'pellet' of who? The legend has to stand alone; even if the reader doesn't read the rest of the paper, he/she should understand what the figure is about.

Table 1 - I believe that the comma used here should be changed by dots. In Italian, the comma might mean a decimal number, but in English, it points to thousands. Please, check this information.

Table 2 - refer to my comment to figure 6. Different response variables to model what?

I look forward to seeing this nice work published soon.
All the best,
Emilio

---

## Round 0.3 · accepted · Accept

Dear Dr. Longo,

You have paid careful attention to the reviewers' comments. The manuscript has been substantially revised in accordance with the comments of the reviewers. If we need to clarify any details required to move the manuscript forward, then our production staff will get in touch with you. Otherwise, a proof will be forthcoming shortly for your review.

Congratulations and thank you for your submission.

The Section Editors added:

> The authority for the species looks incorrect based on WoRMS (year as 1814) so please make sure authors address this in proof stage